# DECORRELATED DOUBLE Q-LEARNING

## ABSTRACT

Q-learning with value function approximation may have the poor performance because of overestimation bias and imprecise estimate. Specifically, overestimation bias is from the maximum operator over noise estimate, which is exaggerated using the estimate of a subsequent state. Inspired by the recent advance of deep reinforcement learning and Double Q-learning, we introduce the decorrelated double Q-learning (D2Q). Specifically, we introduce Q-value function utilizing control variates and the decorrelated regularization to reduce the correlation between value function approximators, which can lead to less biased estimation and low variance. The experimental results on a suite of MuJoCo continuous control tasks demonstrate that our decorrelated double Q-learning can effectively improve the performance.

## 1 INTRODUCTION

Q-learning Watkins & Dayan (1992) as a model free reinforcement learning approach has gained popularity, especially under the advance of deep neural networks Mnih et al. (2013). In general, it combines the neural network approximators with the actor-critic architectures Witten (1977); Konda & Tsitsiklis (1999), which has an actor network to control how the agent behaves and a critic to evaluate how good the action taken is.

The Deep Q-Network (DQN) algorithm Mnih et al. (2013) firstly applied the deep neural network to approximate the action-value function in Q-learning and shown remarkably good and stable results by introducing a target network and Experience Replay buffer to stabilize the training. Lillicrap et al. proposes DDPG Lillicrap et al. (2015), which extends Q-learning to handle continuous action space with target networks. Except the training stability, another issue Q-learning suffered is overestimation bias, which was first investigated in Thrun & Schwartz (1993). Because of the noise in function approximation, the maximum operator in Q-learning can lead to overestimation of state-action values. And, the overestimation property is also observed in deterministic continuous policy control Silver & Lever (2014). In particular, with the imprecise function approximation, the maximization of a noisy value will induce overestimation to the action value function. This inaccuracy could be even worse (e.g. error accumulation) under temporal difference learning Sutton & Barto (1998), in which bootstrapping method is used to update the value function using the estimate of a subsequent state.

Given overestimation bias caused by maximum operator of noise estimate, many methods have been proposed to address this issue. Double Q-learning van Hasselt (2010) mitigates the overestimation effect by introducing two independently critics to estimate the maximum value of a set of stochastic values. Averaged-DQN Anschel et al. (2017) takes the average of previously learned Q-values estimates, which results in a more stable training procedure, as well as reduces approximation error variance in the target values. Recently, Twin Delayed Deep Deterministic Policy Gradients (TD3) Fujimoto et al. (2018) extends the Double Q-learning, by using the minimum of two critics to limit the overestimated bias in actor-critic network. A soft Q-learning algorithm Haarnoja et al. (2018), called soft actor-critic, leverages the similar strategy as TD3, while including the maximum entropy to balance exploration and exploitation. Maxmin Q-learning Lan et al. (2020) proposes the use of an ensembling scheme to handle overestimation bias in Q-Learning.

This work suggests an alternative solution to the overestimation phenomena, called decorrelated double Q-learning, based on reducing the noise estimate in Q-values. On the one hand, we want to make the two value function approximators as independent as possible to mitigate overestima-

tion bias. On the other hand, we should reduce the variance caused by imprecise estimate. Our decorrelated double Q-learning proposes an objective function to minimize the correlation of two critics, and meanwhile reduces the target approximation error variance with control variate methods. Finally, we provide experimental results on MuJoCo games and show significant improvement compared to competitive baselines.

The paper is organized as follows. In Section 2, we introduce reinforcement learning problems, notations and two existed Q-learning variants to address overestimation bias. Then we present our D2Q algorithm in Section 3 and also prove that in the limit, this algorithm converges to the optimal solution. In Section 4 we show the experimental results on MuJoCo continuous control tasks, and compare it to the current state of the art. Some related work and discussion is presented in Section 5 and finally Section 6 concludes the paper.

## 2 BACKGROUND

In this section, we introduce the reinforcement learning problems and Q-learning, as well as notions that will be used in the following sections.

### 2.1 PROBLEM SETTING AND NOTATIONS

We consider the model-free reinforcement learning problem (i.e. optimal policy existed) with sequential interactions between an agent and its environment Sutton & Barto (1998) in order to maximize a cumulative return. At every time step $t$, the agent selects an action $a_t$ in the state $s_t$ according its policy and receives a scalar reward $r_t(s_t, a_t)$, and then transit to the next state $s_{t+1}$. The problem is modeled as Markov decision process (MDP) with tuple: $(\mathcal{S}, \mathcal{A}, p(s_0), p(s_{t+1}|s_t, a_t), r(s_t, a_t), \gamma)$. Here, $\mathcal{S}$ and $\mathcal{A}$ indicate the state and action space respectively, $p(s_0)$ is the initial state distribution. $p(s_{t+1}|s_t, a_t)$ is the state transition probability to $s_{t+1}$ given the current state $s_t$ and action $a_t$, $r(s_t, a_t)$ is reward from the environment after the agent taking action $a_t$ in state $s_t$ and $\gamma$ is discount factor, which is necessary to decay the future rewards ensuring finite returns. We model the agent's behavior with $\pi_\theta(a|s)$, which is a parametric distribution from a neural network.

Suppose we have the finite length trajectory while the agent interacting with the environment. The return under the policy $\pi$ for a trajectory $\tau = (s_t, a_t)_{t=0}^{T}$

$$J(\theta) = \mathbb{E}_{\tau \sim \pi_\theta(\tau)}[r(\tau)] = \mathbb{E}_{\tau \sim \pi_\theta(\tau)}[R_0^T]$$

$$= \mathbb{E}_{\tau \sim \pi_\theta(\tau)}[\sum_{t=0}^{T} \gamma^t r(s_t, a_t)] \tag{1}$$

where $\pi_\theta(\tau)$ denotes the distribution of trajectories,

$$p(\tau) = \pi(s_0, a_0, s_1, ..., s_T, a_T)$$

$$= p(s_0) \prod_{t=0}^{T} \pi_\theta(a_t|s_t) p(s_{t+1}|s_t, a_t) \tag{2}$$

The goal of reinforcement learning is to learn a policy $\pi$ which can maximize the expected returns

$$\theta = \arg\max_\theta J(\theta) = \arg\max \mathbb{E}_{\tau \sim \pi_\theta(\tau)}[R_0^T] \tag{3}$$

The action-value function describes what the expected return of the agent is in state $s$ and action $a$ under the policy $\pi$. The advantage of action value function is to make actions explicit, so we can select actions even in the model-free environment. After taking an action $a_t$ in state $s_t$ and thereafter following policy $\pi$, the action value function is formatted as:

$$Q^\pi(s_t, a_t) = \mathbb{E}_{s_i \sim p_\pi, a_i \sim \pi}[R_t|s_t, a_t] = \mathbb{E}_{s_i \sim p_\pi, a_i \sim \pi}[\sum_{i=t}^{T} \gamma^{(i-t)} r(s_i, a_i)|s_t, a_t] \tag{4}$$

To get the optimal value function, we can use the maximum over actions, denoted as $Q^*(s_t, a_t) = \max_\pi Q^\pi(s_t, a_t)$, and the corresponding optimal policy $\pi$ can be easily derived by $\pi^*(s) \in \arg\max_{a_t} Q^*(s_t, a_t)$.

## 2.2 Q-LEARNING

Q-learning, as an off-policy RL algorithm, has been extensively studied since it was proposed Watkins & Dayan (1992). Suppose we use neural network parametrized by $\theta^Q$ to approximate Q-value in the continuous environment. To update Q-value function, we minimize the follow loss:

$$L(\theta^Q) = \mathbb{E}_{s_i \sim p_\pi, a_i \sim \pi}[(Q(s_t, a_t; \theta^Q) - y_t)^2] \tag{5}$$

where $y_t = r(s_t, a_t) + \gamma \max_{a_{t+1}} Q(s_{t+1}, a_{t+1}; \theta^Q)$ is from Bellman equation, and its action $a_{t+1}$ is taken from frozen policy network (actor) to stabilizing the learning. In actor-critic methods, the policy $\pi : \mathcal{S} \mapsto \mathcal{A}$, known as the actor with parameters $\theta^\pi$, can be updated through the chain rule in the deterministic policy gradient algorithm Silver & Lever (2014)

$$\nabla J(\theta^\pi) = \mathbb{E}_{s \sim p_\pi}[\nabla_a Q(s, a; \theta^Q)|_{a=\pi(s;\theta^\pi)} \nabla_{\theta^\pi}(\pi(s; \theta^\pi))] \tag{6}$$

where $Q(s, a)$ is the expected return while taking action $a$ in state $s$, and following $\pi$ after.

One issue has attracted great attention is overestimation bias, which may exacerbate the situation into a more significant bias over the following updates if left unchecked. Moreover, an inaccurate value estimate may lead to poor policy updates. To address it, Double Q-learning van Hasselt (2010) use two independent critics $q_1(s_t, a_t)$ and $q_2(s_t, a_t)$, where policy selection uses a different critic network than value estimation

$$q_1(s_t, a_t) = r(s_t, a_t) + \gamma q_2(s_{t+1}, \arg\max_{a_{t+1}} q_1(s_{t+1}, a_{t+1}; \theta^{q_1}); \theta^{q_2})$$

$$q_2(s_t, a_t) = r(s_t, a_t) + \gamma q_1(s_{t+1}, \arg\max_{a_{t+1}} q_2(s_{t+1}, a_{t+1}; \theta^{q_2}); \theta^{q_1})$$

Recently, TD3 Fujimoto et al. (2018) uses the similar two q-value functions, but taking the minimum of them below:

$$y_t = r(s_t, a_t) + \gamma \min(q_1(s_{t+1}, \pi(s_{t+1})), q_2(s_{t+1}, \pi(s_{t+1}))) \tag{7}$$

Then the same square loss in Eq. 5 can be used to learn model parameters.

## 3 DECORRELATED DOUBLE Q-LEARNING

In this section, we present Decorrelated Double Q-learning (D2Q) for continuous action control with attempt to address overestimation bias. Similar to Double Q-learning, we use two q-value functions to approximate $Q(s_t, a_t)$. Our main contribution is to borrow the idea from control variates to decorrelate these two value functions, which can further reduce the overestimation risk.

### 3.1 Q-VALUE FUNCTION

Suppose we have two approximators $q_1(s_t, a_t)$ and $q_2(s_t, a_t)$, D2Q uses the weighted difference of double q-value functions to approximate the action-value function at $(s_t, a_t)$. Thus, we define Q-value as following:

$$Q(s_t, a_t) = q_1(s_t, a_t) - \beta(q_2(s_t, a_t) - E(q_2(s_t, a_t))) \tag{8}$$

where $q_2(s_t, a_t) - E(q_2(s_t, a_t))$ is to model the noise in state $s_t$ and action $a_t$, and $\beta$ is the correlation coefficient of $q_1(s_t, a_t)$ and $q_2(s_t, a_t)$. To understand the expectation $E(q_2(s_t, a_t))$, it is the average over all possible runs. Thus, the weighted difference between $q_1(s_t, a_t)$ and $q_2(s_t, a_t)$ attempts to reduce the variance and remove the noise effects in Q-learning.

To update $q_1$ and $q_2$, we minimize the following loss:

$$L(\theta^Q) = \mathbb{E}_{s_i \sim p_\pi, a_i \sim \pi}[(q_1(s_t, a_t; \theta^{q_1}) - y_t)^2] + \mathbb{E}_{s_i \sim p_\pi, a_i \sim \pi}[(q_2(s_t, a_t; \theta^{q_2}) - y_t)^2]$$
$$+ \lambda \mathbb{E}_{s_i \sim p_\pi, a_i \sim \pi}[corr(q_1(s_t, a_t; \theta^{q_1}), q_2(s_t, a_t; \theta^{q_2}))]^2 \tag{9}$$

where $\theta^Q = \{\theta^{q_1}, \theta^{q_2}\}$, and $y_t$ can be defined as

$$y_t = r(s_t, a_t) + \gamma Q(s_{t+1}, a_{t+1}) \tag{10}$$

where $Q(s_{t+1}, a_{t+1})$ is the action-value function defined in Eq. 8 to decorrelate $q_1(s_{t+1}, a_{t+1})$ and $q_2(s_{t+1}, a_{t+1})$, which are both from the frozen target networks. In addition, we want these two q-value functions as independent as possible. Thus, we introduce $corr(q_1(s_t, a_t; \theta^{q_1}), q_2(s_t, a_t; \theta^{q_1}))$, which measures similarity between these two q-value approximators. In the experiment, our method using Eq. 10 can get good results on Halfcheetah, but it did not perform well on other MuJoCo tasks.

To stabilize the target value, we take the minimum of $Q(s_{t+1}, a_{t+1})$ and $q_2(s_{t+1}, a_{t+1})$ in Eq. 10 as TD3 Fujimoto et al. (2018). Then, it gives the target update of D2Q algorithm below

$$y_t = r(s_t, a_t) + \gamma \min(Q(s_{t+1}, a_{t+1}), q_2(s_{t+1}, a_{t+1})) \tag{11}$$

And the action $a_{t+1}$ is from policy $a_{t+1} = \pi(s_{t+1}; \theta^\pi)$, which can take a similar policy gradient as in Eq. 6. Our D2Q leverages the parametric actor-critic algorithm, which maintains two q-value approixmators and a single actor. Thus, the loss in Eq. 9 tries to minimize the three terms below, as

$$corr(q_1(s_t, a_t; \theta^{q_1}), q_2(s_t, a_t; \theta^{q_2})) \to 0$$
$$q_1(s_t, a_t; \theta^{q_1}) \to y_t$$
$$q_2(s_t, a_t; \theta^{q_2}) \to y_t$$

At each time step, we update the pair of critics towards the minimum target value in Eq. 11, while reducing the correlation between them. The purposes that we introduce control variate $q_2(s_t, a_t)$ are following: (1) Since we use $q_2(s_t, a_t) - E(q_2(s_t, a_t))$ to model noise, if there is no noise, such that $q_2(s_t, a_t) - E(q_2(s_t, a_t)) = 0$, then we have $y_t = r(s_t, a_t) + min(Q^\pi(s_t, a_t), q_2(s_t, a_t)) = r(s_t, a_t) + min(q_1(s_t, a_t), q_2(s_t, a_t))$ via Eq. 11, which is exactly the same as TD3. (2) In fact, because of the noise in value estimate, we have $q_2(s_t, a_t) - E(q_2(s_t, a_t)) \neq 0$. The purpose we introduce $q_2(s_t, a_t)$ is to mitigate overestimate bias in Q-learning. The control variate introduced by $q_2(s_t, a_t)$ will reduce the variance of $Q(s_t, a_t)$ to stabilize the learning of value function.

**Convergence analysis**: we claim that our D2Q algorithm is to converge the optimal in the finite MDP settings. There is existed theorem in Jaakkola et al. (1994), given the random process $\{\Delta_t\}$ taking value in $\mathbb{R}^n$ and defined as

$$\Delta_{t+1}(s_t, a_t) = (1 - \alpha_t(s_t, a_t))\Delta_t(s_t, a_t) + \alpha_t(s_t, a_t)F_t(s_t, a_t) \tag{12}$$

Then $\Delta_t$ converges to zero with probability 1 under the following assumptions:

1. $0 < \alpha_t < 1, \sum_t \alpha_t(x) = \infty$ and $\sum_t \alpha_t^2(x) < \infty$

2. $||E[F_t(x)|\mathcal{F}_t]||_W \leq \gamma||\Delta_t||_W + c_t$ with $0 < \gamma < 1$ and $c_t \xrightarrow{p} 0 = 1$

3. $var[F_t(x)|\mathcal{F}_t] \leq C(1 + ||\Delta_t||_W^2)$ for $C > 0$

where $\mathcal{F}_t$ is a sequence of increasing $\sigma$-field such that $\alpha_t(s_t, a_t)$ and $\Delta_t$ are $\mathcal{F}_t$ measurable for $t = 1, 2, ....$

Based on the theorem above, we provide sketch of proof which borrows heavily from the proof of convergence of Double Q-learning and TD3 as below: Firstly, the learning rate $\alpha_t$ satisfies the condition 1. Secondly, variance of $r(s_t, a_t)$ is limit, so condition 3 holds. Finally, we will prove that condition 2 holds below.

$$\Delta_{t+1}(s_t, a_t) = (1 - \alpha_t(s_t, a_t))(Q(s_t, a_t) - Q^*(s_t, a_t))$$
$$+ \alpha_t(s_t, a_t)\big(r_t + \gamma \min(Q(s_t, a_t), q_2(s_t, a_t)) - Q^*(s_t, a_t)\big)$$
$$= (1 - \alpha_t(s_t, a_t))\Delta_t(s_t, a_t) + \alpha_t(s_t, a_t)F_t(s_t, a_t) \tag{13}$$

where $F_t(s_t, a_t)$ is defined as:

$$F_t(s_t, a_t) = r_t + \gamma \min(Q(s_t, a_t), q_2(s_t, a_t)) - Q^*(s_t, a_t)$$
$$= r_t + \gamma \min(Q(s_t, a_t), q_2(s_t, a_t)) - Q^*(s_t, a_t) + \gamma Q(s_t, a_t) - \gamma Q(s_t, a_t)$$
$$= r_t + \gamma Q(s_t, a_t) - Q^*(s_t, a_t) + \gamma \min(Q(s_t, a_t), q_2(s_t, a_t)) - \gamma Q(s_t, a_t)$$
$$= F_t^Q(s_t, a_t) + c_t \tag{14}$$

Since we have $E[F_t^Q(s_t, a_t)|\mathcal{F}_t] \leq \gamma||\Delta_t||$ under Q-learning, so the condition 2 holds. Then we need to prove $c_t = \min(Q(s_t, a_t), q_2(s_t, a_t)) - Q(s_t, a_t)$ converges to 0 with probability 1.

$$
\begin{aligned}
&\min(Q(s_t, a_t), q_2(s_t, a_t)) - Q(s_t, a_t) \\
=&\min(Q(s_t, a_t), q_2(s_t, a_t)) - q_2(s_t, a_t) + q_2(s_t, a_t) - Q(s_t, a_t) \\
=&\min(Q(s_t, a_t) - q_2(s_t, a_t), 0) - (Q(s_t, a_t) - q_2(s_t, a_t)) \\
=&\min(q_1(s_t, a_t) - q_2(s_t, a_t) - \beta(q_2(s_t, a_t) - E(q_2(s_t, a_t))), 0) \\
&+ q_1(s_t, a_t) - q_2(s_t, a_t) - \beta(q_2(s_t, a_t) - E(q_2(s_t, a_t)))
\end{aligned}
\tag{15}
$$

Suppose there exists very small $\delta_1$ and $\delta_2$, such that $|q_1(s_t, a_t) - q_2(s_t, a_t)| \leq \delta_1$ and $|q_2(s_t, a_t) - E(q_2(s_t, a_t))| \leq \delta_2$, then we have

$$
\begin{aligned}
&\min(Q(s_t, a_t), q_2(s_t, a_t)) - Q(s_t, a_t) \\
\leq&2(|q_1(s_t, a_t) - q_2(s_t, a_t)| + \beta|q_2(s_t, a_t) - E(q_2(s_t, a_t))|) \\
=&2(\delta_1 + \beta\delta_2) < 4\delta
\end{aligned}
\tag{16}
$$

where $\delta = \max(\delta_1, \delta_2)$. Note that $\exists \delta_1, |q_1(s_t, a_t) - q_2(s_t, a_t)| \leq \delta_1$ holds because $\Delta_t(q_1, q_2) = |q_1(s_t, a_t) - q_2(s_t, a_t)|$ converges to zero. According Eq. 9, both $q_1(s_t, a_t)$ and $q_2(s_t, a_t)$ are updated with following

$$
q_{t+1}(s_t, a_t) = q_t(s_t, a_t) + \alpha_t(s_t, a_t)(y_t - q_t(s_t, a_t))
\tag{17}
$$

Then we have $\Delta_{t+1}(q_1, q_2) = \Delta_t(q_1, q_2) - \alpha_t(s_t, a_t)\Delta_t(q_1, q_2) = (1 - \alpha_t(s_t, a_t))\Delta_t(q_1, q_2)$ converges to 0 as the learning rate satisfies $0 < \alpha_t(s_t, a_t) < 1$.

### 3.2 CORRELATION COEFFICIENT

The purpose we introduce $corr(q_1(s_t, a_t), q_2(s_t, a_t))$ in Eq. 9 is to reduce the correlation between two value approximators $q_1$ and $q_2$. In other words, we hope $q_1(s_t, a_t)$ and $q_2(s_t, a_t)$ to be as independent as possible. In this paper, we define $corr(q_1, q_2)$ as:

$$
corr(q_1(s_t, a_t), q_2(s_t, a_t)) = cosine(f_{q_1}(s_t, a_t), f_{q_2}(s_t, a_t))
$$

where $cosine(a, b)$ is the cosine similarity between two vectors $a$ and $b$. $f_q(s_t, a_t)$ is the vector representation of the last hidden layer in the value approximator $q(s_t, a_t)$. In other words, we constrain the hidden representation learned from $q_1(s_t, a_t)$ and $q_2(s_t, a_t)$ in the loss function, with attempt to make them independent.

According to control variates, the optimal $\beta$ in Eq. 8 is:

$$
\beta = \frac{cov(q_1(s_t, a_t), q_2(s_t, a_t))}{var(q_1(s_t, a_t))}
$$

where $cov$ is the symbol of covariance, and $var$ represents variance. Considering it is difficult to estimate $\beta$ in continuous action space, we take an approximation here. In addition, to reduce the number of hyper parameters, we set $\beta = corr(q_1(s_t, a_t), q_2(s_t, a_t))$ in Eq. 8 to approximate the correlation coefficient of $q_1(s_t, a_t)$ and $q_2(s_t, a_t)$ since it is hard to get covariance in the continuous action space.

### 3.3 ALGORITHM

We summarize our approach in Algorithm. 1. Similar to Double Q-learning, we use the target networks with a slow updating rate to keep stability under temporal difference learning. Our contributions are two folder: (1) introduce the loss to minimize the correlation between two critics, which can make $q_1(s_t, a_t)$ and $q_2(s_t, a_t)$ as random as possible, and then effectively reduce the overestimation risk; (2) add control variates to reduce variance in the learning procedure.

## 4 EXPERIMENTAL RESULTS

In this section, we evaluate our method on the suite of MuJoCo continuous control tasks. We downloaded the OpenAI Gym environment, and used the MuJoCo v2 version of all tasks to test our

---

**Algorithm 1** Decorrelated Double Q-learning

---

Initialize a pair of critic networks $q_1(s, a; \theta^{q_1})$, $q_2(s, a; \theta^{q_2})$ and actor $\pi(s; \theta^{\pi})$ with weights $\theta^Q = \{\theta^{q_1}, \theta^{q_2}\}$ and $\theta^{\pi}$

Initialize corresponding target networks for both critics and actor $\theta^{Q'}$ and $\theta^{\pi'}$;

Initialize the total number of episodes $N$, batch size and the replay buffer $\boldsymbol{R}$

Initialize the coefficient $\lambda$ in Eq. 9

Initialize the updating rate $\tau$ for target networks

**for** episode = 1 **to** $N$ **do**

    Receive initial observation state $s_0$ from the environment

    **for** $t = 0$ **to** $T$ **do**

        Select action according to $a_t = \pi(s_t; \theta^{\pi}) + \epsilon$, $\epsilon \sim \mathcal{N}(0, \sigma)$

        Execute action $a_t$ and receive reward $r_t$, $done$, and further observe new state $s_{t+1}$

        Push the tuple $(s_t, a_t, r_t, done, s_{t+1})$ into $\boldsymbol{R}$

        //sample from replay buffer

        Sample a batch of $D = (s_t, a_t, r_t, done, s_{t+1})$ from $\boldsymbol{R}$

        $a_{t+1} = \pi(s_{t+1}; \theta^{\pi}) + \epsilon$ with clip, $\epsilon \sim \mathcal{N}(0, \tilde{\sigma})$

        Compute $Q(s_t, a_t)$ with target critic networks according to Eq. 8

        Compute target value $y_t$ via Eq. 11

        Update critics $q_1$ and $q_2$ by minimizing $\mathcal{L}(\theta^Q)$ in Eq. 9

        Update actor $a = \pi(s; \theta^{\pi})$ by maximizing $Q(s_t, a_t)$ value in Eq. 8

    **end for**

    Update the target critics $\theta^{Q'} = (1 - \tau)\theta^{Q'} + \tau\theta^Q$

    Update the target actor $\theta^{\pi'} = (1 - \tau)\theta^{\pi'} + \tau\theta^{\pi}$

**end for**

Return parameters $\theta = \{\theta^Q, \theta^{\pi}\}$.

---

method. We compared our approach against the state of the art off-policy continuous control algorithms, including DDPG, SAC and TD3. Since SAC requires the well-tuned hyperparameters to get the maximum reward across different tasks, we used the existed results from its training logs published by its authors. To obtain consistent results, we use the author's implementation for TD3 and DDPG. In practice, while we minimize the loss in Eq. 9, we constrain $\beta \in (0, 1)$. In addition, we add Gaussian noise to action selected by the target policy in Eq. 11. Specifically, the target policy adds noise as $a_{t+1} = \pi(s_{t+1}; \theta^{\pi}) + \epsilon$, where $\epsilon = clip(\mathcal{N}(0, \sigma), -c, c)$ with $c = 0.5$.

Without other specification, we use the same parameters below for all environments. The deep architecture for both actor and critic uses the same networks as TD3 Fujimoto et al. (2018), with hidden layers [400, 300, 300]. Note that the actor adds the noise $\mathcal{N}(0, 0.1)$ to its action space to enhance exploration and the critic networks have two Q-functions $q_1(s, a)$ and $q_2(s, a)$. The minibatch size is 100, and both network parameters are updated with Adam using the learning rate $10^{-3}$. In addition, we also use target networks including the pair of critics and a single actor to improve the performance as in DDPG and TD3. The target policy is smoothed by adding Gaussian noise $\mathcal{N}(0, 0.2)$ as in TD3, and both target networks are updated with $\tau = 0.005$. We set the balance weight $\lambda = 2$ for all tasks except Walker2d which we set $\lambda = 10$. In addition, the off-policy algorithm uses the replay buffer $\boldsymbol{R}$ with size $10^6$ for all experiments.

We run each task for 1 million time steps and evaluate it every 5000 time steps with no exploration noise. We repeat each task 5 times with random seeds and get its mean and standard deviation respectively. And we report our evaluation results by averaging the returns with window size 10. The evaluation curves are shown in Figures 1, 2 and 3. Our D2Q consistently achieves much better performance than TD3 on most continuous control tasks, including InvertedDoublePendulum, Walker2d, Ant, Halfcheetah and Hopper environments. Other methods such as TD3 perform well on one task Reacher, but perform poorly on other tasks compared to our algorithm.

We also evaluated our approach on high dimensional continuous action space task. The Humanoid-v2 has 376 dimensional state space and 17 dimensional action space. In the task, we set the learning rate on Humanoid to be $3 \times 10^{-4}$, and compared to DDPG and TD3. The result in Figure 1(b) demonstrates that our performance on this task is on a par with TD3.

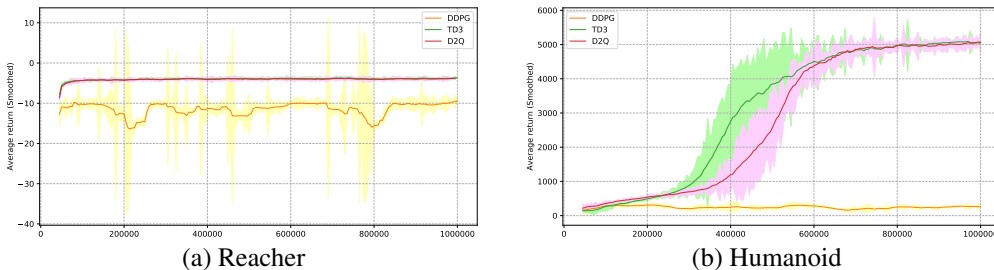

Figure 1: The learning curves with exploration noise on Reacher and Humanoid environments. The shaded region represents the standard deviation of the average evaluation over nearby windows with size 10. On the MuJoCo tasks, our D2Q algorithm yields competitive results, compared to TD3 and DDPG.

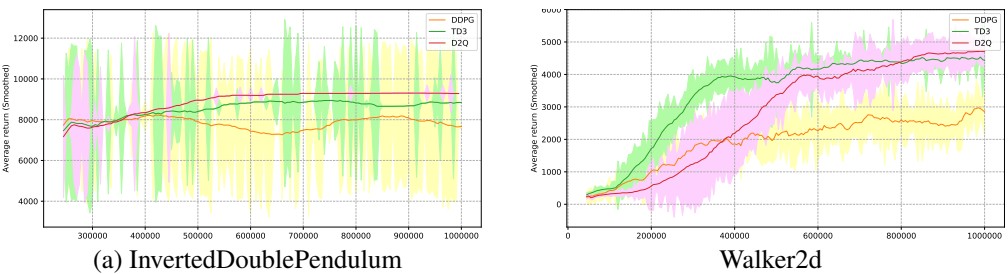

Figure 2: The learning curves with exploration noise on the InvertedDoublePendulum and Walker2d environments. The shaded region represents the standard deviation of the average evaluation over nearby windows with size 10. Our D2Q algorithm yields competitive results, compared to TD3 and DDPG.

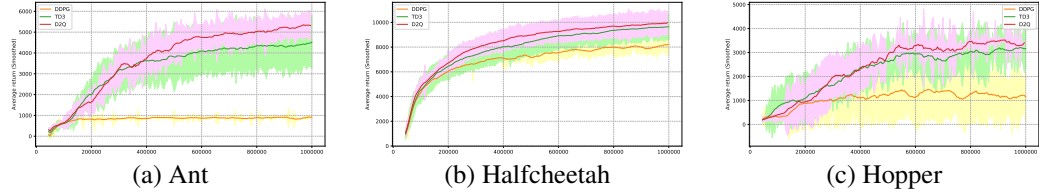

Figure 3: The learning curves with exploration noise on the Ant, Halfcheetah and Hopper environments. The shaded region represents the standard deviation of the average evaluation over nearby windows with size 10. Our D2Q algorithm yields significantly better results, compared to TD3 and DDPG.

The quantitative results over 5 trials are presented in Table 1. Compared to SAC Haarnoja et al. (2018), our approach shows better performance with lower variance given the same size of training samples. It demonstrates that our approach can yield competitive results, compared to TD3 and DDPG. Specifically, our D2Q method outperforms all other algorithms with much low variance on Ant, HalfCheetah, InvertedDoublePendulum and Walker2d. In the Hopper task, our method achieve maximum reward competitive with the best methods such as TD3, with comparable variance.

## 5 RELATED WORK

Q-learning can suffer overestimation bias because it uses the maximum to estimate the maximum expected value. To address the overestimation issue Thrun & Schwartz (1993) in Q-learning, many approaches have been proposed to avoid the maximization operator of a noisy value estimate. Delayed Q-learning Strehl et al. (2006) tries to find $\epsilon$-optimal policy, which determines how frequent to update state-action function. However, it can suffer from overestimation bias, although it guarantees to converge in polynomial time. Double Q-learning van Hasselt (2010) introduces two indepen-

Table 1: Comparison of Max Average Return over 5 trials of 1 million samples. The maximum value is marked bold for each task. $\pm$ corresponds to a single standard deviation over trials.

| Environments | Methods | | | | |
|---|---|---|---|---|---|
| | D2Q | TD3 | SAC | DDPG | PPO |
| HalfCheetah | $\mathbf{9958.3 \pm 935.70}$ | $9636.95 \pm 859.06$ | $8895.96 \pm 3316.5$ | 8577.29 | 1795.43 |
| Hopper | $\mathbf{3364.34 \pm 583.72}$ | $3223.75 \pm 514.2$ | $2100.67 \pm 1051.6$ | 2020.46 | 2164.70 |
| Walker2d | $\mathbf{4727.20 \pm 444.71}$ | $4582.82 \pm 525.60$ | $3475.15 \pm 1508.71$ | 1843.85 | 3317.69 |
| Ant | $\mathbf{5264.69 \pm 632.90}$ | $4373.44 \pm 1000.33$ | $3250.49 \pm 1157.94$ | 1005.30 | 1082.20 |
| Reacher | $-3.78 \pm 0.32$ | $\mathbf{-3.6 \pm 0.56}$ | NA | -6.51 | -6.18 |
| InvPendulum | $\mathbf{1000 \pm 0.0}$ | $1000 \pm 0.0$ | NA | 1000 | 1000 |
| InvDoublePendulum | $\mathbf{9200.6 \pm 186.22}$ | $8911.04 \pm 750.58$ | NA | $7741.28 \pm 2195.87$ | 8977.94 |

dently trained critics to mitigate the overestimation effect. Averaged-DQN Anschel et al. (2017) takes the average of previously learned Q-values estimates, which results in a more stable training procedure, as well as reduces approximation error variance in the target values. A clipped Double Q-learning called TD3 Fujimoto et al. (2018) extends the deterministic policy gradient Silver & Lever (2014); Lillicrap et al. (2015) to address overestimation bias. In particular, TD3 uses the minimum of two independent critics to approximate the value function suffering from overestimation. Soft actor critic Haarnoja et al. (2018) takes a similar approach as TD3, but with better exploration with maximum entropy method. Maxmin Q-learning Lan et al. (2020) extends Double Q-learning and TD3 to multiple critics to handle overestimation bias and variance.

Another side effect of consistent overestimation Thrun & Schwartz (1993) in Q-learning is that the accumulated error of temporal difference Sutton & Barto (1998) can cause high variance. To reduce the variance, there are two popular approaches: baseline and actor-critic methods Witten (1977); Konda & Tsitsiklis (1999). In policy gradient, we can minus baseline in Q-value function to reduce variance without bias. Further, the advantage actor-critic ($A^2C$) Mnih et al. (2016) introduces the average value to each state, and leverages the difference between value function and the average to update the policy parameters. Schulman et al proposed the generalized advantage value estimation Schulman et al. (2016), which considered the whole episode with an exponentially-weighted estimator of the advantage function that is analogous to $TD(\lambda)$ to substantially reduce the variance of policy gradient estimates at the cost of some bias.

From another point of view, baseline and actor-critic methods can be categories into control variate methods Greensmith et al. (2001). Greensmith et al. analyze the two additive control variate methods theoretically including baseline and actor-critic method to reduce the variance of performance gradient estimates in reinforcement learning problems. Interpolated policy gradient (IPG) Gu et al. (2017) based on control variate methods merges on- and off-policy updates to reduce variance for deep reinforcement learning. Motivated by the Stein's identity, Liu et al. introduce more flexible and general action-dependent baseline functions Liu et al. (2018) by extending the previous control variate methods used in REINFORCE and advantage actor-critic. In this paper, we present a novel variant of Double Q-learning to constrain possible overestimation. We limit the correlation between the pair of q-value functions, and also introduce the control variates to reduce variance and improve performance.

## 6 CONCLUSION

In this paper, we propose the Decorrelated Double Q-learning approach for off-policy value-based reinforcement learning. We use a pair of critics for value estimate, but we introduce a regularization term into the loss function to decorrelate these two approixmators. While minimizing the loss function, it constrains the two q-value functions to be as independent as possible. In addition, considering the overestimation derived from the maximum operator over positive noise, we leverage control variates to reduce variance and stabilize the learning procedure. The experimental results on a suite of challenging tasks in the continuous control environment demonstrate our approach yields on par or better performance than competitive baselines. Although we leverage control variates in our q-value function, we approximate the correlation coefficient with a simple strategy based on the similarity of these two q-functions. In the future work, we will consider a better estimation of correlation coefficient in control variate method.

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

## A APPENDIX

We add additional experiments on how our model will perform by varying $\lambda$ in this Appendix. We set $\lambda = [1, 2, 5, 10]$ respectively to run 1 Million steps and evaluate its performance every 5000 steps, while keeping all other parameters same.

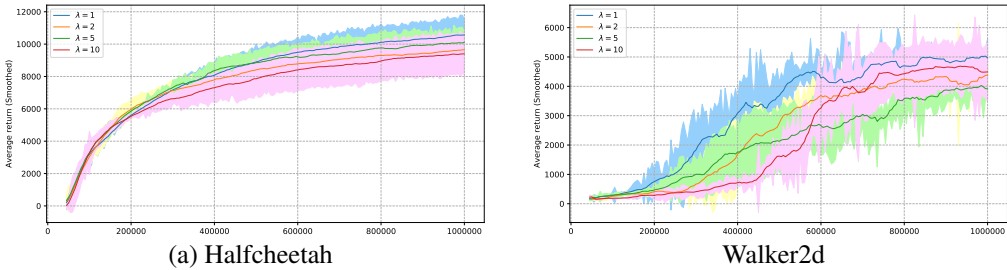

(a) Halfcheetah          Walker2d

Figure 4: The figures show how our method will perform while adjusting parameter $\lambda$. The shaded region represents the standard deviation of the average evaluation over nearby windows with size 10.

