# OpenReview forum: "Decorrelated Double Q-learning"
_ICLR.cc/2021/Conference — Reject_

### Official Review · AnonReviewer3 · 2020-10-26
**Improving double Q-learning with decorrelated regularization and control variates**

**Rating:** 4
**Confidence:** 3

**Review:**

The proposed "decorrelated double Q-learning" algorithm combines a few techniques to improve the performance of model-free RL, including control variates for reducing variance, decorrelated regularization for reducing bias, and a technique from TD3 for stabilizing learning.

Overall, the ideas of this work are interesting and bring some insights for tackling the overestimation issue of Q-learning. Empirically, the proposed method shows some improvements over the existing ones. However, a few major concerns are as follows.

- The theoretical analysis of convergence seems hand-waving and confuses me. For example, does the analysis only apply to the tabular case? (The authors don't seem to state this explicitly.) How does Eq. (17) follow from Eq. (9) (are we missing the gradient of the decorrelated regularization term)?

- All experimental results are about reward vs. iteration curves, which are not convincing or insightful enough. For example, is there empirical evidence showing that the proposed algorithm does learn two decorrelated critics?

- The structure of Section 3 may need some adjustment. In particular, in the current version, the formal definition of the correlation term (Sec 3.2) and the description of the full algorithm itself (Sec 3.3) appear after the convergence analysis of the algorithm (Sec 3.1), which looks weird.

Based on the above comments, I think substantial improvements are needed for publication of this work.

---

### Official Review · AnonReviewer1 · 2020-10-26
**Strange theory and unconvincing experiments**

**Rating:** 3
**Confidence:** 3

**Review:**

Summary
The paper suggests an improvement over double-Q learning by applying the control variates technique to the target Q, in the form of $(q1 - \beta (q2 - E(q2))$ (eqn (8)). To minimize the variance, it suggests minimizing the correlation between $q1$ and $q2$. In addition, it applies the TD3 trick. The resulting algorithm, D2Q, outperforms DDPG and competes with TD3.

Recommendation
I hope I haven’t misunderstood this paper, but I’ve found neither the theory nor the experiment convincing. Therefore I recommend a rejection.

Strengths
The proposed algorithm is simple and straightforward to use.

Weaknesses
1. Theory
(a) Minimizing the variance of eqn (8) requires maximizing the correlation between q1 and q2. If they are independent, what’s the point of including q2? Check out https://en.wikipedia.org/wiki/Control_variates
(b) $E(q2))$ is the “average over all possible runs”. It’s unclear how it’s calculated. Maybe run a few identical RL experiments with different random seeds, just to get $E(q2)$? Feels wasteful to me.
(c) Why would minimizing the squared cosine between last layer feature vectors lead to minimum correlation? If the feature for q2 is obtained from that of q1 through a deterministic 90º rotation, wouldn’t that result in a zero cosine but really strong correlation?
(d) Why is it ok to ignore $var(q1)$ while computing $\beta$? No theory is given here.

2. Experiments
In Fig 1 - 3, D2Q sometimes outperforms and sometimes underperforms TD3. Because these two algorithms are so similar, I can’t tell whether the comparison is statistically significant.

Other feedbacks
Please address questions raised above. Perform additional experiments to make the paper more convincing.

---

### Official Review · AnonReviewer2 · 2020-10-26
**Proof is hard to go through**

**Rating:** 3
**Confidence:** 4

**Review:**

The paper proposes a method that modifies double Q-learning by eliminating a linearly correlated part of one Q. I am not familiar with the proof of Double Q-learning and TD3, and thus find the proof of this paper hard to read as it omits the majority of proof by claiming it is similar to the proof of the aforementioned two algorithms. To name some of the part that confused me while reading: what is the definition of F^Q_t and c_t in (14)? Why does a small delta_2 exist in (16)? Why does Delta_t converge to zero as claimed in the line after (16)? Why is the randomness of s_{t+1} not mentioned in the subscripts of E's in (5) and (9)? etc. Therefore, I suggest the author(s) write a thorough proof and put it in the appendix to make the convergence analysis readable. I am also curious how the de-correlation term helps to improve the convergence in the analysis as it is the main contribution of this paper. Besides,  double Q-learning and TD are mostly used in function approximations. I wonder if the analysis can extend to some simple case of parameterized Q functions, e.g. linear approximations. The experiment part looks good to me as it compares D2Q with several sota algorithms and get satisfying results.

---

### Official Review · AnonReviewer4 · 2020-10-28
**This paper designs a variant of Double Q-learning to deal with overestimation and high variance, and provides experiment results to compare the new algorithm with previous ones.**

**Rating:** 5
**Confidence:** 4

**Review:**

Overall, I vote for rejecting. The core idea of the new algorithm looks interesting, but this paper does not provide convincing evidence theoretically and empirically.

Pros:

1. This paper introduces a new variant of Double Q-learning to reduce the overestimation risk and reduce variance.
2. In part of the experiments, the new algorithm demonstrates better performance than previous ones.

Cons:

1. This paper does not provides enough insightful intuition and theoretical guarantees on the design of the new algorithm. There should be more explanation and evidence to help reader understand why for instance, the definition of Q-value in equation (8) makes sense.
2. This paper is not well-written. There are missing references links, typos and missing definitions of some notations. For example, in the second paragraph of introduction, “Lillicrap et al.” is just text and not linked to a paper in the references and there are many other cases like this. Typos can be seen in many places as well, such as in the first paragraph of section 3.3, it should be ‘two-fold’ instead of ‘two folder’; in the last paragraph of section 5, ‘categories’ should be ‘categorised’. Moreover, in the convergence analysis, the meaning of many notations are not explained at all (e.g. $\alpha_t$). The author only says “we provide sketch of proof which borrows heavily from the proof of convergence of Double Q-learning and TD3”, but without the definitions of the notations, the completeness of the paper is greatly undermined.

---

### Public Comment · ~Qiang_He1 · 2020-11-11
**Good idea, lack of citation.**

Your idea looks good. However, the comparison experiments lack similar methods which reduce the estimation error.
Recommend the author compare with the following method:  [1]Li Z, Hou X. Mixing Update Q-value for Deep Reinforcement Learning[C]//2019 International Joint Conference on Neural Networks (IJCNN). IEEE, 2019: 1-6. [2]He Q, Hou X. Reducing Estimation Bias via Weighted Delayed Deep Deterministic Policy Gradient[J]. arXiv preprint arXiv:2006.12622, 2020.

---

### Decision · Program_Chairs · 2021-01-07
**Final Decision**

**Decision:**

Reject

**Comment:**

This paper investigates some variants of the double Q-learning algorithm and develops theoretical guarantees. In particular, it focuses on how to reduce the correlation between the two trajectories employed in the double Q-learning strategy, in the hope of rigorously addressing the overestimation bias issue that arises due to the max operator in Q-learning. However, the reviewers point out that the proofs are hard to parse (and often hand-waving with important details omitted). The experimental results are also not convincing enough.